# Characteristics and conflicts of interest at Food and Drug Administration Gastrointestinal Drug Advisory Committee meetings

Rishad Khan[1], Karam Elsolh[2,3], Nikko Gimpaya[2], Michael A. Scaffidi[2,4], Rishi Bansal[2,3], Samir C. Grover[1,2,5]*

1 Department of Medicine, University of Toronto, Toronto, Canada, 2 Division of Gastroenterology, St. Michael's Hospital, Toronto, Canada, 3 Michael G. DeGroote School of Medicine, McMaster University, Hamilton, Canada, 4 School of Medicine, Queen's University, Kingston, Canada, 5 Li Ka Shing Knowledge Institute, St. Michael's Hospital, Toronto, Canada

* samir.grover@utoronto.ca

**Data Availability Statement:** The data underlying the results presented in the study are available from S1 File as well as on the public domain of the United States Food and Drug Administration

## Abstract

### Introduction

The United States Food and Drug Administration (FDA) Gastrointestinal Drug Advisory Committee (GIDAC) is involved in gastrointestinal drug application reviews. Characteristics and conflicts of interest (COI) in GIDAC meetings are not well described. This study analyzed FDA GIDAC meetings and characteristics that predict recommendations.

### Methods

In this cross-sectional study, all publicly available GIDAC meetings where proposed medications were voted on were included. Data were collected regarding indications, medication sponsor, primary efficacy studies, and voting member characteristics (e.g. committee membership, COI). Univariate analyses were conducted at per-meeting and per-vote levels to assess for predictors of committee recommendation and individual votes respectively.

### Results

Thirty-four meetings with 476 individual votes from 1998–2018 were included. Twenty-three (68%) proposals were recommended for approval and 25 (74%) received FDA approval. Most proposals involved >1 primary study (n = 27, 79%). At least one voting member had a COI in 24 (71%) of 34 meetings. Twelve (35%) meetings had at least one sponsor COI. Among 476 individual votes, 74 (15.5%) involved a COI, with 33 (6.9%) sponsor COI. COI decreased significantly over time, with fewer COI in 2006–2010, 2011–2015, and 2016–2020 compared to 1996–2000 and 2001–2005 (p<0.01). There were no significant associations between pre-defined predictors, including COI, and committee level recommendations or individual votes (p>0.05 for all univariate analyses).

Gastrointestinal Drug Advisory Committee site: https://www.fda.gov/advisory-committees/human-drug-advisory-committees/gastrointestinal-drugs-advisory-committee.

**Funding:** The authors received no specific funding for this work.

**Competing interests:** Rishad Khan has received research grants from Abbvie and Ferring Pharmaceuticals and research funding from Pendopharm. Samir C. Grover has received research grants and personal fees from Takeda, education grants from Janssen, and has equity in Volo Healthcare. All other authors have no relevant disclosures. This does not alter our adherence to PLOS ONE policies on sharing data and materials.

**Abbreviations:** FDA, Food and Drug Administration; GIDAC, Gastrointestinal Drug Advisory Committee; GI, gastrointestinal; COI, conflict of interest; USD, United States Dollar; IQR, interquartile range.

## Conclusions

The GIDAC reviewed 34 proposals from 1998–2018. The majority were recommended for approval and later approved by the FDA, highlighting the GIDAC's prominence in the regulatory process. COI are present among GIDAC panelists but decreasing over time and not associated with recommendations.

## Introduction

The United States Food and Drug Administration (FDA) Gastrointestinal Drug Advisory Committee (GIDAC) plays an important role in the evaluation and approval of new gastrointestinal (GI) drug applications [1]. Little is known, however, about the characteristics of GIDAC meetings. With calls for more transparent and impartial processes in medication approval pathways [2], a systematic understanding of the GIDAC's role is needed.

Advisory committees are typically convened when the FDA requires additional clinical expertise to review a drug application, assess medications with controversial risk-benefit profiles, or independently review data for a new product [3]. While committees are strictly advisory, the high concordance between committee recommendations and the FDA's final actions suggest a prominent role in determining approvals [4]. From 2008 to 2012, the GIDAC recommended approval for 77% of reviewed products, with the FDA later approving 83% of the same products [1]. Despite the committee's presumed importance, the features of GIDAC meetings and factors that influence recommendations, including the prevalence of conflicts of interest (COI), are unknown.

COI are common in other areas of gastroenterology practice. While these relationships between the medical and corporate pharmaceutical companies are necessary for innovation and research [5, 6], they can affect the objectivity of clinical and scientific recommendations and decision making [7–9]. The prevalence of COI has been demonstrated in GI society guidelines [10–14] and been shown to correlate with increased prescribing of expensive biologic medications [15]. COI among FDA panelists has been studied broadly across all committees and specifically for oncology drug applications [16–18], but are not well described for GIDAC panelists.

To address this gap in the understanding of the GIDAC its role in the approval process, we analyzed FDA GIDAC meetings and evaluated characteristics that predict positive recommendations.

## Materials and methods

We conducted a cross-sectional analysis of all GIDAC meetings available from the FDA website [19]. We did not restrict our search by time. As this study uses publicly available data, it does not qualify as human subjects research, based on the Tri-council Policy Statement on Ethical Conduct for Research Involving Humans and is thus exempt from ethical review [20].

### Sample

Two authors (RK, KE) independently and in duplicate searched all GIDAC meetings and identified meetings in which voting took place. We included meetings in which votes were cast on questions regarding 1) whether to recommend approval for a medication for a specific indication, 2) whether a medication had demonstrated safety and efficacy, or 3) whether a

medication had a favorable risk-benefit profile. We excluded meetings if they voted on broad topics that did not concern a specific clinical indication (e.g. voting on a class of drugs) or did not concern approval of a medication (e.g. voting on necessity of post-marketing safety studies).

We counted a single proposed indication for one medication as a single meeting. If there were votes on multiple medications and/or multiple indications for one medication, we counted each indication for each medication as a single meeting. For example, if GIDAC panelists voted on two indications each for two medications each on a single day, we would have considered this as four unique meetings.

## Data collection

We extracted data primarily from meeting transcripts. We also used other documents that were available on the FDA website alongside meeting transcripts, including meeting minute summaries, presentation slides, medical officer review documents, and COI waivers [19].

Two authors (RK, KE) collected the following data from meeting transcripts independently and in duplicate: medication name, proposed indication, country of the medication sponsor, type of application (new drug or biological licensing), and whether the majority of the committee voted for approval of a medication for the proposed indication. We used the medication application number to determine if the FDA approved the medication for the proposed indication after the meeting. A third author (SCG) reviewed the data collected and discrepancies were resolved by consensus.

For each proposed indication, we also collected data on the underlying primary studies used as evidence of efficacy and safety, including number of trials reviewed at meetings, and for the pivotal trial(s), study design, sample size, study phase, type of primary endpoint (clinical or surrogate), study result, and whether the study's corresponding author had any COI with the medication sponsor.

We then identified all individual voting members at each meeting. For each vote, we collected the following data on the individual casting the vote: role within the GIDAC, educational degree (if applicable), whether they had any COI, and if so, if the COI were with the medication sponsor or a competitor. We also identified the value of COI in United States Dollar (USD) and classified them as general payments (e.g. consulting, speaker's fees, food and travel), research payments, or equity, in keeping with Centres for Medicare and Medicaid Services definitions of industry payments [21]. Data regarding COI were available in the meeting transcript as part of the executive secretary's statement.

## Analysis

We calculated descriptive statistics with proportions and median with interquartile range (IQR) for categorical and continuous variables respectively. We presented summary data on a per-meeting and per-vote basis. For COI, we presented data stratified by five time periods: 1996–2000, 2001–2005, 2006–2010, 2011–2015, 2016–2020.

We conducted univariable logistic regression to evaluate the relationship between meeting level factors and a positive recommendation for the medication. Predictor variables were dichotomized and included disease subgroup (inflammatory bowel disease vs. other), number of clinical trials reviewed in the meeting (1 vs. >1), sample size of the pivotal trial (<500 vs. ≥500), endpoint (clinical vs. surrogate), efficacy (whether the primary endpoint was achieved or not), medication sponsor (US-based vs. other), type of application (new drug vs. biological licensing). We also included predictor variables for three types of COI: COI present for the corresponding author of the pivotal trial, COI present for any GIDAC member, and COI

present for any GIDAC member with the sponsor of the medication. We intended on including the type of study (randomized trial vs. other) and study phase (phase 3 vs. other) for the pivotal trial but were limited by small subgroups.

To evaluate the relationship between characteristics of GIDAC members and voting pattern, we conducted univariable logistic regression with the following dichotomous predictors: committee membership (chair or standing member vs temporary member), degree (medical vs. other), any COI, COI with the medication sponsor, and COI with a competitor of the medication sponsor. Statistical significance was set at $p < 0.005$ for the logistic regression of meeting level votes and $p < 0.001$ for the logistic regression of GIDAC member voting, as we applied a Bonferroni adjustment for multiple comparisons. We did not use multiple regression analyses to avoid over-fitting this small data set.

We conducted post-hoc analyses with chi-squared testing to determine if there were differences with respect to proportion of votes with COI between time periods (1996–2000, 2001–2005, 2006–2010, 2011–2015, 2016–2020). We followed this analysis with pairwise chi-squared tests between the five different 5-year time periods. All analyses were performed with SPSS version 26 (IBM corporation, Armonk, New York, United States).

## Results

There were 42 available meetings on FDA GIDAC webpage, ranging from 1998 to 2018. Of all meetings, 34 involved voting on proposed indications and comprised our study sample. There were 476 votes cast in the 34 meetings. All primary data are available in S1 File.

### Meeting, study, and vote characteristics

Meetings most commonly involved medications for the management of inflammatory bowel disease (n = 14, 41%), functional GI disease (5, 15%) and liver disease (3, 9%). The majority of medication sponsors were US-based pharmaceutical companies (18, 53%). New drug applications (23, 68%) were more common than biological licensing applications (11, 32%) (**Table 1**).

The committee recommended approval for 23 (68%) indications and the FDA approved 25 (74%) indications, with 21 (62%) approvals for proposed indications and 4 (12%) modified approvals (**Table 2**).

The committee reviewed one primary study in seven (21%) meetings, and more than one trial at 27 (79%) meetings (**Table 3**). The median sample size of pivotal trials was 263 (IQR 80–672). The majority of pivotal trials were phase 3 (33, 97%) randomized controlled trials (30, 88%), had a clinical primary outcome (30, 88%), achieved the primary outcome (30, 88%), and had corresponding authors with COI (23, 68%).

Among the 476 votes cast at 34 meetings, there were 310 (65.1%) yes votes, 161 (33.8%) no votes, and 5 (1.1%) abstentions (**Table 4**). With respect to panelists who cast votes, they were most commonly a temporary committee member/expert consultant (222, 46.6%) or standing GIDAC member (106, 22.3%). There were 34 (7.1%) committee chairs, corresponding to one chair per meeting. Most panelists had medical degrees (360, 75.6%).

### Conflicts of interest

There was at least one voting member with a COI in 24 (71%) of the 34 meetings. Twelve (35%) meetings had at least one sponsor COI and 18 (53%) had at least one competitor COI. Of the 476 individual votes, 74 (15.5%) involved a committee member with COI, comprised of 33 (6.9%) COI with sponsors and 41 (8.6%) with competitors. Ten of the 74 COI were among committee chairs. All members who had COI received waivers and were able to participate in voting. When stratified by time period, the number of votes that involved COI as a proportion

**Table 1. Meeting characteristics.**

| Characteristic | Number of meetings |
|---|---|
| **n (%)** | **N = 34** |
| **Disease group** | |
| Inflammatory bowel disease | 14 (41) |
| Functional gastrointestinal disease | 5 (15) |
| Liver disease | 3 (9) |
| Other [†] | 12 (35) |
| **Sponsor** | |
| United States | 18 (53) |
| United Kingdom | 7 (21) |
| Europe | 6 (17) |
| Other [‡] | 3 (9) |
| **Type of application** | |
| New drug | 23 (68) |
| Biological licensing | 11 (32) |
| **Approval** | |
| Committee recommended approval | 23 (68) |
| FDA provided approval | 25 (74) |
| Approval for proposed indication | 21 (62) |
| Approval for modified indication | 4 (12) |
| **Conflict of interest** | |
| One member with COI | 24 (71) |
| With sponsor | 12 (35) |
| With competitor | 18 (53) |

[†] Heartburn, chemotherapy associated nausea and vomiting, short bowel syndrome, Barrett's esophagus, post-operative ileus, exocrine pancreatic insufficiency, and neonatal hemolysis

[‡] Japan, Canada

*FDA*, Food and Drug Administration; *IQR*, interquartile range

of all votes cast in that time period were as follows: 24 (32.0%) in 1996–2000, 25 (32.8%) in 2001–2005, 5 (7.8%) from 2006–2010, 10 (6.5%) from 2011–2015, and 11 (10.2%) from 2016–2020 (**Fig 1**). Among the 74 COI, 31 (42%) were equity, 12 (16%) were general payments, three (4%) were research payments, and 28 (38%) were not stated. Payment values were most commonly 10,000–49,999 USD (28, 38%), not stated (22, 30%), or less than 10,000 USD (16, 21%) (**Fig 2**).

There was a significant difference with respect to the proportion of votes with COI between the different time periods (p<0.01). On pairwise comparisons, the 1996–2000 and 2001–2005 time periods each had significantly more votes with COI compared with the 2006–2010, 2011–2015, and 2016–2020 time periods (p<0.01). There were no differences when comparing the

**Table 2. Committee recommendations and subsequent FDA approvals.**

| Applications, n (%) | | FDA decision | | |
|---|---|---|---|---|
| N = 34 | | Yes–proposed indication | Yes–modified indication | No |
| GIDAC recommendation | Yes | 20 (59) | 2 (6) | 1 (3) |
| | No | 1 (3) | 2 (6) | 8 (23) |

*FDA*, Food and Drug Administration; *GIDAC*, Gastrointestinal Drug Advisory Committee

**Table 3. Characteristics of studies evaluated for drug applications.**

| Study characteristic | Number of meetings |
|---|---|
| n (%) | N = 34 |
| **Number of trials evaluated in meeting** | |
| 1 | 7 (21) |
| >1 | 27 (79) |
| **Sample size [†], median (IQR)** | 263 (80–672) |
| **Type of study** | |
| Randomized controlled trial | 30 (88) |
| Other [‡] | 4 (12) |
| **Study Phase** | |
| Phase 3 | 33 (97) |
| Phase 2 | 1 (1) |
| **Primary endpoint** | |
| Clinical endpoint | 30 (88) |
| Surrogate endpoint | 4 (12) |
| **Efficacy** | |
| Primary endpoint achieved | 29 (85) |
| Primary endpoint not achieved | 5 (15) |
| **COI present for corresponding author** | 23 (68) |

[†] For meetings which evaluated multiple trials, the sample size from the pivotal trial, or when there were multiple pivotal trials, the largest trial was used

[‡] Includes two non-randomized trials and two single arms trials

*COI*, conflict of interest

**Table 4. Medication approval voting details.**

| Characteristic | Number of votes |
|---|---|
| n (%) | N = 476 |
| **GIDAC membership status** | |
| Chair | 34 (7.1) |
| Standing member | 106 (22.3) |
| Temporary member/expert consultant | 222 (46.6) |
| Patient representative | 27 (5.7) |
| Consumer representative | 28 (5.9) |
| Other committee [†] | 57 (12.0) |
| Federal employee | 2 (0.4) |
| **Degree** | |
| MD/MBBS/DO | 360 (75.6) |
| PhD | 46 (9.7) |
| Pharm D | 18 (3.8) |
| Not stated | 40 (8.4) |
| Other [‡] | 12 (2.5) |
| **COI** | 74 (15.5) |
| Direct (with sponsor) | 33 (6.9) |
| Competitor | 41 (8.6) |

[†] From meetings which involved other committees, such as the Pediatric Drug Advisory Committee

[‡] Includes BA, MPH, and JD

*GIDAC*, Gastrointestinal Drug Advisory Committee; *COI*, conflict of interest

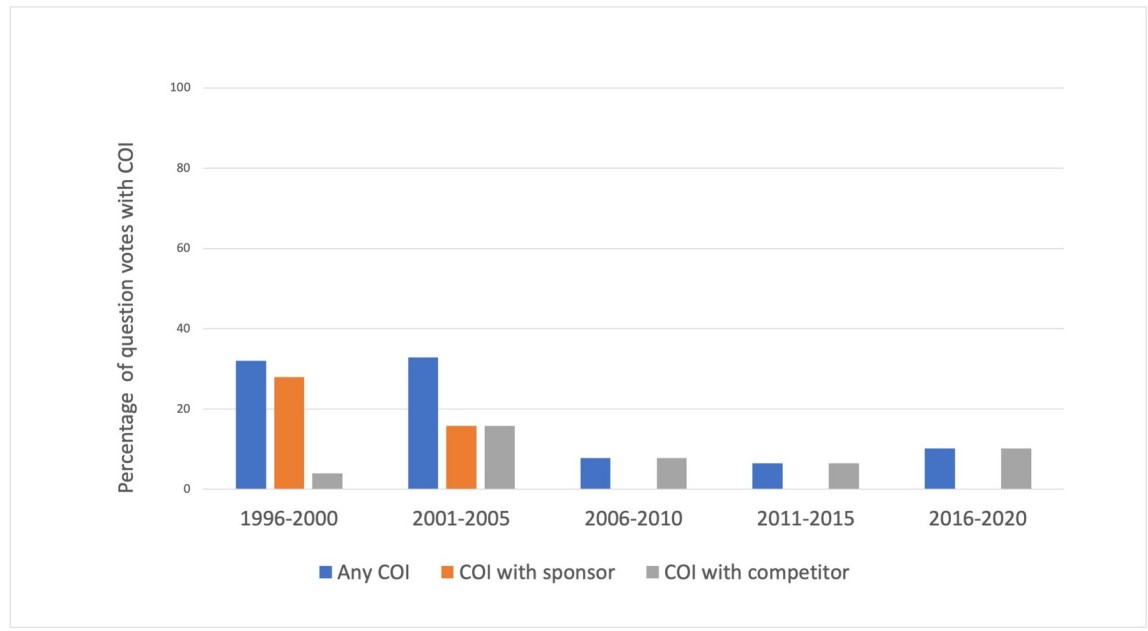

**Fig 1. Percentage of votes at Gastrointestinal Drug Advisory Committee meetings with member conflict of interest, stratified by 5-year time intervals.**

1996–2000 and the 2001–2005 time periods, or when comparing the 2006–2010, 2011–2015, and 2016–2020 time periods (p>0.05).

## Predictors of committee recommendation and votes of approval

There were no significant associations between our *a priori* defined predictor variables and recommendations. For committee-level recommendation, none of disease type, number of

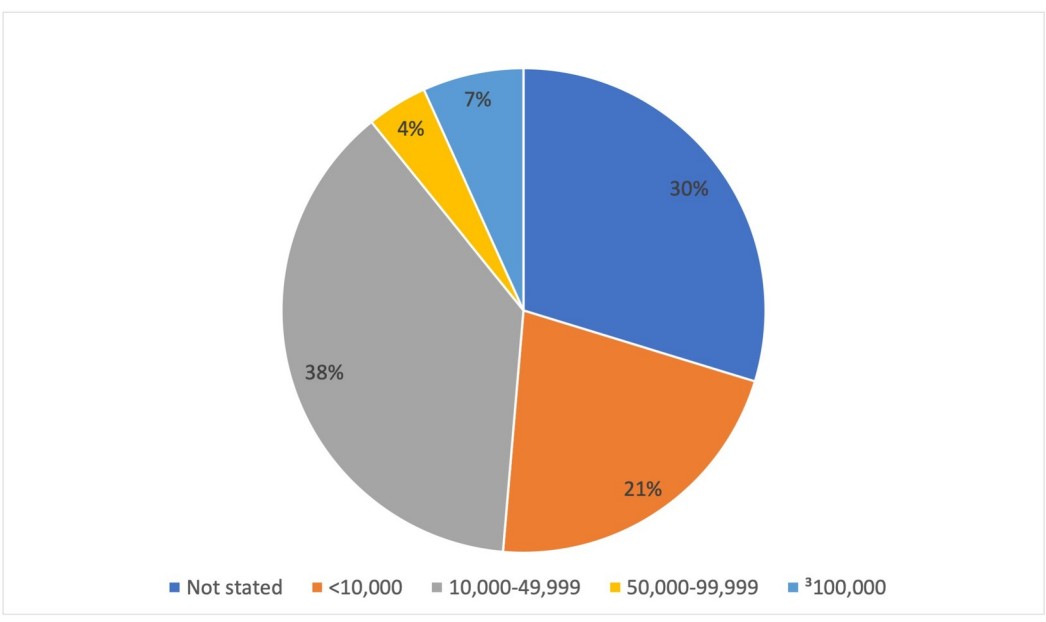

**Fig 2. Value of payments for the 74 votes that involved conflicts of interest.**

**Table 5. Predictors of receiving a GIDAC recommendation.**

| Variable | GIDAC recommendation | |
|---|---|---|
| | OR (95% CI) | *P* value |
| Disease (inflammatory bowel disease vs. other) | 4.91 (0.86–27.88) | 0.07 |
| Number of trials (1 vs. >1) | 1.25 (0.20–7.75) | 0.88 |
| Sample size (<500 vs. ≥500) | 0.55 (0.11–2.73) | 0.46 |
| Endpoint (clinical vs. surrogate) | 2.33 (0.28–19.24) | 0.43 |
| Efficacy (primary endpoint achieved vs. not) | 3.94 (0.55–28.12) | 0.12 |
| Sponsor (US vs. other) | 2.72 (0.62–12.04) | 0.19 |
| Type of application (new drug vs. biological licensing | 0.35 (0.06–1.98) | 0.23 |
| COI present for trial corresponding author | 4.32 (0.92–20.27) | 0.06 |
| COI present for any committee member | 0.86 (0.17–4.23) | 0.85 |
| COI present for any committee member with sponsor | 0.92 (0.22–3.92) | 0.914 |

*GIDAC*, Gastrointestinal Drug Advisory Committee; *OR*, odds ratio.

trials, sample size, endpoint type, efficacy, origin of sponsor, type of application, COI of trial author, COI for any committee member, and COI for any committee member with the sponsor were associated with the GIDAC recommending a medication for approval for the proposed indication (**Table 5**). Additionally, GIDAC membership status, degree, any COI, COI with sponsor, and COI with competitor were not associated with an individual vote of approval (**Table 6**).

## Discussion

We report the first study to systematically analyze FDA GIDAC meetings using both meeting- and vote-level data. Of the 34 meetings in our sample, we found that the majority of these received recommendations of approval from the both the GIDAC and FDA. Additionally, most proposals involved review of more than one primary study, with a median sample size of 263 and the use of clinical, rather than surrogate outcomes. In terms of COI, approximately 15% of voting members reported industry payments, wherein these payments decreased over time. Finally, we did not find any significant predictors for GIDAC recommendations or individual votes.

There are several findings that warrant discussion. First, the GIDAC recommended approval for 68% of proposals, with the FDA later approving 74% of those same proposals, in keeping with previous analyses highlighting the crucial role of advisory committees in determining approval [1, 4]. Second, the number of studies used to demonstrate efficacy, dichotomised to 1 or >1, was not associated with recommending a medication for approval. This

**Table 6. Predictors of votes in favour of approval.**

| Variable | Vote of approval | |
|---|---|---|
| | OR (95% CI) | *P* value |
| GIDAC membership (chair or standing vs. temporary) | 1.50 (0.98–2.30) | 0.06 |
| Degree (medical vs. other) | 1.03 (0.66–1.59) | 0.90 |
| Any COI | 1.06 (0.63–1.79) | 0.83 |
| COI with sponsor | 1.18 (0.56–2.39) | 0.66 |
| COI with competitor | 0.94 (0.47–1.91) | 0.87 |

*COI*, conflict of interest

finding is discordant with FDA's recommendation that two or more well-controlled studies, each convincing on its own, be conducted to demonstrate efficacy [22]. This result may be due to the relatively few patients available for recruitment in the clinical areas certain drugs are aimed at, such as short bowel syndrome, Barrett's esophagus, and hepatic encephalopathy. Flexibility in methodological requirements for rare diseases has been shown for oncologic drug approvals [23].

One area that merits further exploration is that of COI among GIDAC members. Previous studies have established the high prevalence COI are common in other areas in gastroenterology, with previous studies showing a high prevalence of industry payments related to GI guidelines and point-of-care articles on IBD [15, 24, 25]. In this study, 12 (35%) of 34 meetings had at least one member who had a COI with a sponsor and 33 (6.9%) of 476 votes involved COI with a sponsor. Though the FDA has been criticized for poor oversight and regulation of advisory committee interactions with pharmaceutical companies [26, 27], there have been considerable efforts to curb undue industry influence. Prominent examples include the 2002 revision of FDA disclosure guidelines [28], the 2007 FDA Amendments Act [29], and open-access publication of committee members' financial ties [30]. The 2002 FDA guideline revision mandated that any voting members who received waivers for COI would need to disclose whether the COI involved the sponsor or competitor of the product being voted on [28]. In the 2007 Amendments Act, the FDA chose to limit the waivers (for COI) granted to only cases where the waiver is necessary to afford the committee essential expertise [29]. Additionally, voting members would generally be disqualified if they, their spouse, or their minor child, had more than $50,000 in financial interests that could affect the meeting, regardless of need for their expertise [29]. Such changes have likely reduced COI burden, as demonstrated by decreasing COI over time among GIDAC members and no COI with sponsors from 2006 onwards.

In addition to a temporal decrease in COI prevalence, the COI that did exist in our study were not associated with pro-sponsor voting in meetings. Two previous studies, however, which evaluated FDA drug advisory meetings from 2001 to 2004 [17] and 1997–2011 respectively, both found an association between COI and pro-sponsor voting [16]. Our results may discordant for two reasons. First, we only included GIDAC meetings. In studies that include all meetings, it is possible that committees that review and approve more drugs, such as the cardiovascular and renal drug committee and the oncologic drug advisory committee [16], can skew the analysis. Additionally, we had an expanded timeline, with meetings spanning from 1998 to 2018. The implementation of policy changes during this time may have mitigated industry influence. Studies with more updated timelines, including an analysis of Oncology Drug Advisory Meetings from 2000 to 2014 [31] and another of COI among all FDA advisory committee members from 2008–2014 [18], showed results similar to ours, in that there were no associations between COI and voting.

We note the study limitations. COI data are based on self-declarations and may be underestimated [32]. Use of a systematically maintained database, such as the Centre for Medicare and Medicaid Services Open Payments Database [33], may identify additional undisclosed COI. We also had a relatively small sample size and imprecision for some analyses, as evidence by several instances of wide confidence intervals. Our analysis was further limited by an inability to evaluate for factors such as rarity of certain diseases and voting member expertise. Finally, our use of dichotomization of predictor variables may have introduced bias into our analysis [34].

## Conclusion

The GIDAC reviewed 34 proposals for unique indications from 1998 to 2018. The majority received a recommendation for approval and subsequent approval from the FDA, highlighting

the GIDAC's prominent role in the regulatory process. We did not identify any predictors for meeting-level recommendations or individual votes in favour of approval. COI are present among GIDAC panelists, though they have decreased over time and have not been associated with recommendations or votes. Future studies should include an objective assessment of COI among FDA panelists by using resources such as the Open Payments database and search for predictors of approval in a larger sample of meetings.

## Supporting information

**S1 Checklist.**
(DOCX)

**S1 File. Raw data on drug advisory meeting characteristics, voting member characteristics, and financial conflicts of interest.**
(XLSX)

## Author Contributions

**Conceptualization:** Rishad Khan, Karam Elsolh, Samir C. Grover.

**Data curation:** Rishad Khan, Karam Elsolh.

**Formal analysis:** Rishad Khan, Nikko Gimpaya, Michael A. Scaffidi, Rishi Bansal.

**Investigation:** Rishad Khan, Nikko Gimpaya, Michael A. Scaffidi, Rishi Bansal.

**Methodology:** Rishad Khan, Nikko Gimpaya, Michael A. Scaffidi, Rishi Bansal.

**Project administration:** Samir C. Grover.

**Resources:** Samir C. Grover.

**Software:** Samir C. Grover.

**Supervision:** Samir C. Grover.

**Writing – original draft:** Rishad Khan, Karam Elsolh, Samir C. Grover.

**Writing – review & editing:** Rishad Khan, Karam Elsolh, Nikko Gimpaya, Michael A. Scaffidi, Rishi Bansal, Samir C. Grover.

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
