## [Decision Letter · Decision Letter 0]

10 Feb 2021

PONE-D-20-36649

Characteristics and Conflicts of Interest at Food and Drug Administration Gastrointestinal Drug Advisory Committee Meetings

PLOS ONE

Dear Dr. Khan,

Thank you for submitting your manuscript to PLOS ONE. After careful consideration, we feel that it has merit but does not fully meet PLOS ONE’s publication criteria as it currently stands. Therefore, we invite you to submit a revised version of the manuscript that addresses the points raised during the review process.

We look forward to receiving your revised manuscript.

Kind regards,

Claudio Gentili

Academic Editor

PLOS ONE

"Rishad Khan has received research grants from AbbVie and Ferring Pharmaceuticals and research funding from Pendopharm. Samir C. Grover has received research grants and personal fees from AbbVie and Ferring Pharmaceuticals, personal fees from Takeda, education grants from Janssen, and has equity in Volo Healthcare. All other authors have no relevant disclosures."

Reviewers' comments:

Reviewer's Responses to Questions

**Comments to the Author**

1. Is the manuscript technically sound, and do the data support the conclusions?

Reviewer #1: Yes

Reviewer #2: Yes

2. Has the statistical analysis been performed appropriately and rigorously? 

Reviewer #1: Yes

Reviewer #2: No

3. Have the authors made all data underlying the findings in their manuscript fully available?

Reviewer #1: Yes

Reviewer #2: No

4. Is the manuscript presented in an intelligible fashion and written in standard English?

Reviewer #1: Yes

Reviewer #2: Yes

5. Review Comments to the Author

Reviewer #1: This manuscript investigates conflict-of-interest among members of the FDA’s Gastrointestinal Drug Advisory Committee and whether that affects how the committee votes. The study is well done and the results are not terribly surprising given the previous work on this topic that has been done recently. That said, this is still a topic that continues to deserve to be investigated.

1.I would suggest that the authors look at the number of COI waivers per meeting to see what percent of the people had a level of conflicts than would normally be allowed and how those people voted? Similarly, given the influence that committee chairs can have over the process, it would be interesting to look at whether the chairs had a COI and how they voted.

2.Were there any committee meetings where a majority of the members had a conflict?

3.The authors should describe what level of COI disqualifies someone from sitting on a committee and give more detail about how those criteria changed with the 2002 revision of the disclosure guidelines and the 2007 FDA amendments.

4.Lines 187-189: I'd like to see a bit more detailed analysis of these numbers, i.e., out of x instances where the advisory committee recommended approval, the FDA approved y but did not approve z. Out of x studies where the committee did not recommend approval the FDA did not approve y but approved z. This could be done in the form of a 2x2 table.

5.How were discrepancies between people who extracted data resolved?

6.Line 157: What was the basis for dichotomizing sample size between <500 and ≥500. Also, I think that for the number of clinical trials the authors mean 1 vs >1, not 1 vs 12.

7.In the Methods under the Sample subheading the authors said that they also looked at votes for whether a medication had demonstrated safety and efficacy and whether a medication had a favorable risk-benefit profile. Were there no votes on these two questions at any of the committee meetings?

8.Why didn't the authors use multiple linear regression to examine whether the various factors that they analyzed determined a negative versus positive vote?

Reviewer #2: The study is a potentially relevant contribution, even if, as the authors acknowledge, there are important limitations, and the scope is somewhat narrow.

My major concern regards data sharing. Indeed, the GIDAC meetings are public, so the raw data are accessible. However, to ensure reproducibility, given that the authors made several coding and extraction decisions, curated datasets of all extracted information from the GIDAC meetings for each application and indication should be included, along with all extracted information about the individual voting members. Synthetically, the readers need to have access to the entirety of the data supporting their findings and be able to re-run the analyses, without necessarily having to extract everything ab novo.

This information should be included as supplementary material, for example in several tables, or deposited in a public repository such as the Open Science Framework.

Another concern is the rationale for dichotomizing all predictor variables, particularly continuous ones like sample size? There are many problems with dichotomization in general, see for example https://www.bmj.com/content/332/7549/1080.1

Also, the statistical significance threshold should be corrected for the multiple comparisons performed.

6. PLOS authors have the option to publish the peer review history of their article (what does this mean?). If published, this will include your full peer review and any attached files.

Reviewer #1: **Yes: **Joel Lexchin

Reviewer #2: **Yes: **Ioana Alina Cristea

---

## [Author Response · Author response to Decision Letter 0]

22 Feb 2021

February 23 2021

Dr. Gentili

Dear Dr. Gentili and the members of the editorial board:

Thank you for considering and giving us the opportunity to revise our manuscript titled “Characteristics and Conflicts of Interest at Food and Drug Administration Gastrointestinal Drug Advisory Committee Meetings” (PONE-D-20-36649). We would also like to thank the reviewers for their comments. Our responses are below. 

EDITOR

a. We have made the required formatting changes. 

2. We note that you have indicated that data from this study are available upon request. PLOS only allows data to be available upon request if there are legal or ethical restrictions on sharing data publicly

a. We have included the data as supplementary file 1. 

3. Please confirm that this does not alter your adherence to all PLOS ONE policies on sharing data and materials, by including the following statement: "This does not alter our adherence to PLOS ONE policies on sharing data and materials.” (as detailed online in our guide for authors. Please include your updated Competing Interests statement in your cover letter; we will change the online submission form on your behalf.

a. We have included this statement in our cover page. 

REVIEWER 1

4. I would suggest that the authors look at the number of COI waivers per meeting to see what percent of the people had a level of conflicts than would normally be allowed and how those people voted? Similarly, given the influence that committee chairs can have over the process, it would be interesting to look at whether the chairs had a COI and how they voted.

a. We have stated that all voting members with COI had waivers, and that ten of the chairs had COI. Both of these statements are in the Results section, Conflicts of interest subsection. 

5. Were there any committee meetings where a majority of the members had a conflict?

a. There were no meetings where a majority of the members had a conflict. 

6. The authors should describe what level of COI disqualifies someone from sitting on a committee and give more detail about how those criteria changed with the 2002 revision of the disclosure guidelines and the 2007 FDA amendments.

a. We have added several details regarding this legislature in the section “Discussion”. 

7. Lines 187-189: I'd like to see a bit more detailed analysis of these numbers, i.e., out of x instances where the advisory committee recommended approval, the FDA approved y but did not approve z. Out of x studies where the committee did not recommend approval the FDA did not approve y but approved z. This could be done in the form of a 2x2 table.

a. We have added in a table as suggested, which is now table 2. 

8. How were discrepancies between people who extracted data resolved?

a. We have stated that discrepancies were resolved by consensus in the Methods section, Data collection subsection. 

9. Line 157: What was the basis for dichotomizing sample size between <500 and ≥500. Also, I think that for the number of clinical trials the authors mean 1 vs >1, not 1 vs 12.

a. We agree that the dichotomization of certain variables could have introduced bias into our analysis, and have listed this as a limitation in the Discussion section. 

10. In the Methods under the Sample subheading the authors said that they also looked at votes for whether a medication had demonstrated safety and efficacy and whether a medication had a favorable risk-benefit profile. Were there no votes on these two questions at any of the committee meetings?

a. Meeting members voted on either of the above questions. In the 34 meetings we examined, there were no meetings in which both sets of questions were answered

11. Why didn't the authors use multiple linear regression to examine whether the various factors that they analyzed determined a negative versus positive vote?

a. We did not use multiple regression to avoid overfitting this small data set. We have stated this in the Methods section, Analysis subsection. 

REVIEWER 2

12. My major concern regards data sharing. Indeed, the GIDAC meetings are public, so the raw data are accessible. However, to ensure reproducibility, given that the authors made several coding and extraction decisions, curated datasets of all extracted information from the GIDAC meetings for each application and indication should be included, along with all extracted information about the individual voting members. Synthetically, the readers need to have access to the entirety of the data supporting their findings and be able to re-run the analyses, without necessarily having to extract everything ab novo. This information should be included as supplementary material, for example in several tables, or deposited in a public repository such as the Open Science Framework.

a. We have included the data as a supplementary file. 

13. Another concern is the rationale for dichotomizing all predictor variables, particularly continuous ones like sample size? There are many problems with dichotomization in general, see for example11/19/15 9:25:00 AM

a. We acknowledge the deficiencies with dichotomizing predictor variables. However, given the relatively small sample size and the likely lack of a normal distribution for the continuous variables, we felt it would be more appropriate to dichotomize the variables. Additionally, we based our dichotomization partially on a similar analysis on oncology drug advisory meetings by Tibau et al. (2016). We have added in statement regarding the limitation on data dichotomization in the Discussion, limitations paragraph. 

14. Also, the statistical significance threshold should be corrected for the multiple comparisons performed.

a. We have added in a Bonferroni correction in the Methods section. 

Thank you once again for your consideration of this manuscript. We look forward to your review and comments.

Yours sincerely,

Samir C. Grover, MD, MEd, FRCPC

Division of Gastroenterology

St. Michael's Hospital

---

## [Decision Letter · Decision Letter 1]

7 Apr 2021

PONE-D-20-36649R1

Characteristics and Conflicts of Interest at Food and Drug Administration Gastrointestinal Drug Advisory Committee Meetings

PLOS ONE

Dear Dr. Khan,

Thank you for submitting your manuscript to PLOS ONE. After careful consideration, we feel that it has merit but does not fully meet PLOS ONE’s publication criteria as it currently stands. Therefore, we invite you to submit a revised version of the manuscript that addresses the points raised during the review process.

We look forward to receiving your revised manuscript.

Kind regards,

Claudio Gentili

Academic Editor

PLOS ONE

Journal Requirements:

Reviewers' comments:

Reviewer's Responses to Questions

**Comments to the Author**

1. If the authors have adequately addressed your comments raised in a previous round of review and you feel that this manuscript is now acceptable for publication, you may indicate that here to bypass the “Comments to the Author” section, enter your conflict of interest statement in the “Confidential to Editor” section, and submit your "Accept" recommendation.

Reviewer #1: (No Response)

Reviewer #2: All comments have been addressed

2. Is the manuscript technically sound, and do the data support the conclusions?

Reviewer #1: Yes

Reviewer #2: (No Response)

3. Has the statistical analysis been performed appropriately and rigorously? 

Reviewer #1: Yes

Reviewer #2: (No Response)

4. Have the authors made all data underlying the findings in their manuscript fully available?

Reviewer #1: Yes

Reviewer #2: (No Response)

5. Is the manuscript presented in an intelligible fashion and written in standard English?

Reviewer #1: Yes

Reviewer #2: (No Response)

6. Review Comments to the Author

Reviewer #1: The revisions that the authors have made have dealt with all of my initial comments. There are just some minor copy editing changes that need to be made.

Line 91: Insert “presumed” between “committee’s” and “importance”.

Line 143: It should probably be “trial(s)” rather than “trial”.

Line 165: Do the authors mean >1 instead of 12?

Line 190: The start of this line does not read properly.

Reviewer #2: I cannot access the supplementary file with all the extracted data, but I assume it is an error of the system. Please make sure it is included.

7. PLOS authors have the option to publish the peer review history of their article (what does this mean?). If published, this will include your full peer review and any attached files.

Reviewer #1: **Yes: **Joel Lexchin

Reviewer #2: **Yes: **Ioana A. Cristea

---

## [Author Response · Author response to Decision Letter 1]

13 Apr 2021

April 13 2021

Dr. Gentili

Dear Dr. Gentili and the members of the editorial board:

Thank you for considering and giving us the opportunity to revise our manuscript titled “Characteristics and Conflicts of Interest at Food and Drug Administration Gastrointestinal Drug Advisory Committee Meetings” (PONE-D-20-36649R1). We would also like to thank the reviewers for their comments. Our responses are below. 

REVIEWER 1

Line 91: Insert “presumed” between “committee’s” and “importance”.

Line 143: It should probably be “trial(s)” rather than “trial”.

Line 165: Do the authors mean >1 instead of 12?

Line 190: The start of this line does not read properly.

Response: We have made these changes in the manuscript. 

REVIEWER 2

I cannot access the supplementary file with all the extracted data, but I assume it is an error of the system. Please make sure it is included.

Response: We have uploaded this file as a supplementary XLS file. 

FORMATTING

Response: We have uploaded new figures after using PACE to ensure PLOS requirements. 

Thank you once again for your consideration of this manuscript. We look forward to your review and comments.

Yours sincerely,

Samir C. Grover, MD, MEd, FRCPC

Division of Gastroenterology

St. Michael's Hospital

---

## [Editor Report · Decision Letter 2]

11 May 2021

Characteristics and Conflicts of Interest at Food and Drug Administration Gastrointestinal Drug Advisory Committee Meetings

PONE-D-20-36649R2

Dear Dr. Khan,

We’re pleased to inform you that your manuscript has been judged scientifically suitable for publication and will be formally accepted for publication once it meets all outstanding technical requirements.

Kind regards,

Claudio Gentili

Academic Editor

PLOS ONE
---

## [Editor Report · Acceptance letter]

14 May 2021

PONE-D-20-36649R2 

Characteristics and Conflicts of Interest at Food and Drug Administration Gastrointestinal Drug Advisory Committee Meetings 

Dear Dr. Khan:

I'm pleased to inform you that your manuscript has been deemed suitable for publication in PLOS ONE. Congratulations! Your manuscript is now with our production department. 

Kind regards, 

on behalf of

Professor Claudio Gentili 

Academic Editor

PLOS ONE